# Synthesis and Characterization of Highly Crystalline Vertically Aligned WSe₂ Nanosheets

**Ayrton Sierra-Castillo [1],*** , **Emile Haye [2]**, **Selene Acosta [3]** , **Carla Bittencourt [3]** and **J.-F. Colomer [1]**

1. Research Group on Carbon Nanostructures (CARBONNAGe), University of Namur, 61 Rue de Bruxelles, 5000 Namur, Belgium; jean-francois.colomer@unamur.be
2. Laboratoire Interdisciplinaire de Spectroscopie Electronique (LISE), Namur Institute of Structured Matter (NISM), University of Namur, 61 Rue de Bruxelles, 5000 Namur, Belgium; emile.haye@unamur.be
3. Chimie des Interactions Plasma–Surface (ChIPS), Research Institute for Materials Science and Engineering, Université de Mons, 7000 Mons, Belgium; Selene.ACOSTAMORALES@umons.ac.be (S.A.); Carla.BITTENCOURT@umons.ac.be (C.B.)
* Correspondence: ayrton.sierracastillo@unamur.be; Tel.: +32-492-80-85-43

**Abstract:** Here, we report on the synthesis of tungsten diselenide (WSe₂) nanosheets using an atmospheric pressure chemical vapor deposition technique via the rapid selenization of thin tungsten films. The morphology and the structure, as well as the optical properties, of the so-produced material have been studied using electron microscopies, X-ray photoelectron spectroscopy, photoluminescence, UV–visible and Raman spectroscopies, and X-ray diffraction. These studies confirmed the high crystallinity, quality, purity, and orientation of the WSe₂ nanosheets, in addition to the unexpected presence of mixed phases, instead of only the most thermodynamically stable 2H phase. The synthesized material might be useful for applications such as gas sensing or for hydrogen evolution reaction catalysis.

**Keywords:** tungsten diselenide; nanosheets; chemical vapor deposition

## 1. Introduction

Tungsten diselenide (WSe₂) belongs to the transition-metal dichalcogenides (TMDs) family. The chemical formula of TMDs is $MX_2$, where M is a transition metal (Mo, W, Nb, Ta, etc.) and X is a chalcogenide (S, Se, etc.). The $MX_2$ compounds form layers composed of three atomic planes, namely: one metal atom between two chalcogenide atoms, covalently bonded, and the layers are linked between them by van der Waals forces [1]. Here, this is the case of the material of interest, WSe₂ [2]. Bulk WSe₂ has attracted increasing attention because of its interesting properties, exemplified by its ultralow thermal conductivity at room temperature ($0.05 \text{ W·m}^{-1}\text{·K}^{-1}$) when considering disordered crystals [3]. This bulk material has also been reported to be efficient in different applications, such as catalysts for hydrogen evolution reactions [4,5], or in photovoltaic devices [6]. However, the recent craze for this material comes from the possibility of obtaining TMDs as 2D materials, similar to graphene, but with complementary electronic properties; these are indeed semi-conductors [7]. Actually, it has been reported that the properties of TMDs, such as MoS₂, WS₂, and WSe₂, depend directly on the number of layers in the structure, especially in systems with few layers. [7]. Moreover, monolayer WSe₂ possesses a small band-gap (smaller than monolayer MoS₂), and shows an ambipolar transport phenomenon [8]. Several potential optoelectronic applications have been described using monolayer WSe₂, such as photodetectors [9], light-emitting diodes [10], and solar-energy convertors [11].

The attractive reported properties and potential applications of WSe₂ materials require a well-controlled synthesis, in terms of their structure, including the number of layers, crystallographic



phase composition, or/and film morphology. For this purpose, different synthesis techniques, such as chemical-vapor transport using a sealed ampoule containing W and Se materials under a vacuum and heated at a high temperature [4,12–14], chemical and mechanical exfoliation [15,16], physical techniques (molecular beam epitaxial growth [17], pulsed laser deposition [18], and magnetron sputtering of W in an Se-rich atmosphere [6]), chemical approaches (colloidal method [19,20] and electrodeposition [21]), and atmospheric pressure chemical vapor deposition (APCVD) [22–26], have been used to obtain $WSe_2$. The majority of recent studies are focused on the synthesis of 2D $WSe_2$, with domains of different shapes and sizes. Furthermore, only a few are dedicated to thin films' synthesis with a controlled morphology, orientation, and crystallography, although these characteristics depend on the production method and growth parameters [14,23,27]. We report here on a simple growth strategy using the APCVD technique to obtain vertically-aligned $WSe_2$ nanosheets by rapid selenization. The developed method is an atmospheric pressure system technique that is rapid, scalable, and cost-effective, and has been applied previously for aligned $MoS_2$ nanosheets [28]. The produced samples were characterized using usual techniques such as electron microscopies, photoluminescence (PL), UV–visible and Raman spectroscopies, X-ray diffraction (XRD), and X-ray photoelectron spectroscopy (XPS).

## 2. Materials and Methods

### 2.1. Materials

The commercial products, namely the W target (purity 99.95%) and Se powder (purity 99%), were purchased from Micro to Nano and Alfa Aesar, respectively, and were used as received.

#### 2.1.1. Synthesis of $WSe_2$ Nanosheets

The $WSe_2$ nanosheets were grown via double selenization using an atmospheric pressure CVD technique. A 50-nm thick W film was deposited on a sapphire substrate by direct current magnetron sputtering (sputter current 100 mA), using a commercial sputter deposition system (Quorum Q15T/ES) in an Argon (99.9995%) atmosphere. A pure W target with a 57 mm diameter was used, and the substrates were placed on a rotating holder with a 90 mm target. The pressure of the argon in the deposition chamber was $1 \times 10^{-3}$ mbar. The W film thickness was monitored using a quartz microbalance mounted in the deposition system.

The selenization was performed on a quartz tube (reactor). Firstly, the W film on the sapphire substrate was introduced into the reactor with the Se powder, and this was placed in two predefined zones of the tube (0.350 g were used in each zone) in order to be in the correct temperature zones when the reactor was inserted into the furnace. The Se powder was used without any further purification. The reactor was flushed for one hour to remove (reduce) oxygen, using a 0.475 L min$^{-1}$ argon flow outside of the furnace, prior to the selenization process. After that, the reactor was put inside of the furnace. As in a typical selenization process (Figure S1), the Se powder was placed into two temperature zones, one part at 40 °C and the other at 850 °C (in total 0.70 g), along with the W film sample, which was placed at 850 °C with the argon flow, into the reactor. A 0.150 L-min$^{-1}$ $H_2$ flow was inserted to help with the reaction. The presence of a strong reducer is indeed mandatory for the selenization reaction, compared with the sulfurization reaction. In the first selenization step of 30 min, the Se powder was moved to the 850 °C zone. The optimized second selenization step was performed by inserting the quartz tube into the hot zone of the furnace, such that the Se powder placed at 40 °C reached the 400 °C temperature zone; during an additional 30 min, the sample remained in the good temperature zone of 850 °C (moving only a few centimeters). After the reaction, the $H_2$ flow was stopped, and the quartz tube was removed from the reactor and was cooled with the argon flow for 1 h.

#### 2.1.2. Characterization Techniques

Scanning and transmission electron microscopy (SEM and TEM, respectively) analyses were conducted on a JEOL 7500F microscope operating at 15 kV and on a TECNAI T20 microscope

working under 200 kV, respectively. The WSe$_2$ nanosheets were examined using Raman spectroscopy, photoluminescence (PL), and UV–visible spectroscopy. The Raman and PL spectra were obtained by using a micro-Raman system (Senterra Bruker Optik GmbH) with a 3 cm$^{-1}$ resolution, using a laser excitation laser source (532 nm wavelength), and a laser power of 2 and 5 mW, respectively. X-ray diffraction (XRD) was used to characterize the samples using a Panalytical X'Pert PRO diffractometer (comprising Cu K$\alpha$ radiation, Bragg–Brentano geometry, a sealed tube operated at 45 mA 30 kV, and a X'Celerator linear detector). The chemical composition of the WSe$_2$ nanosheets was studied with X-ray photoelectron spectroscopy (XPS) using an Escalab 250i Thermo Fisher Scientific$^{TM}$ instrument (consisting of a monochromatic Al K$\alpha$ X-ray source and a hemispherical deflector analyzer working at a constant pass energy).

## 3. Results and Discussion

The sample morphology, resulting from the selenization of W film deposited on the sapphire substrate, was examined by SEM (Figure 1). The observations show that the sample is composed of well-distributed platelets over the entire substrate surfaces. These platelets are nanosheets with a well-defined shape exhibiting sharp edges. Two main contrasts, clearer or darker, can be observed in the lower magnification image (Figure 1a), and can be related to the two orientations of nanosheets to the substrate, perpendicular or parallel, respectively. The higher magnification image (Figure 1b, enlargement of a clearer contrast area) indicates that the WSe$_2$ nanosheets with a thickness between 40–50 nm are preferably grown vertically (perpendicularly to the substrate).

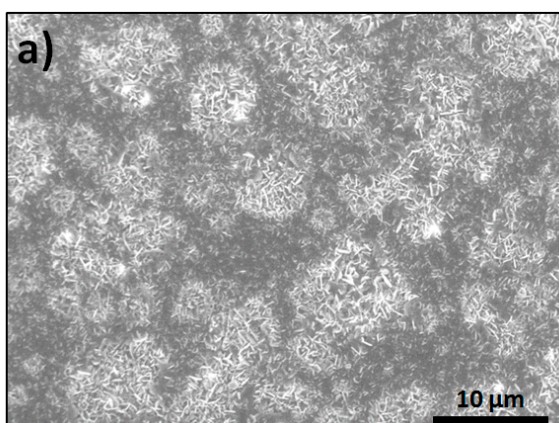
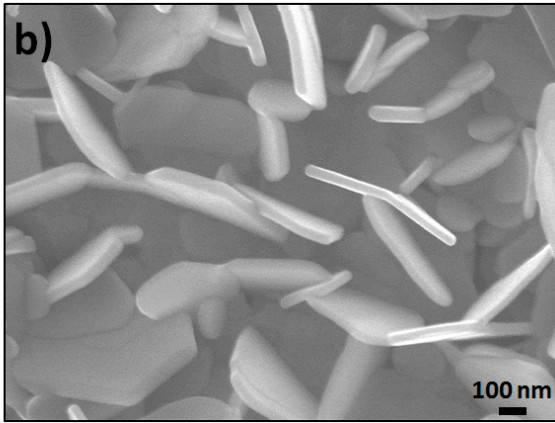

**Figure 1.** SEM images of tungsten diselenide (WSe$_2$) with (**a**) low and (**b**) high magnification (×2500 and ×60,000, respectively).

The morphology of the nanosheets was investigated using TEM. The nanosheets were removed from the substrate by scratching, and were collected on a holey-carbon copper grid. The TEM analyses confirm the SEM observations above. A TEM image of a nanosheet with a large edge is shown in Figure 2a, which exhibits dark fringes related to the (002) planes with an interlayer spacing of 0.65 nm, valued according to the theoretical spacing of 0.648 nm. The sample was also characterized using selected-area electron diffraction. The shape of the electron diffraction pattern, given in Figure 2b, is explained by the selected area, which includes many nanosheets, without preferential orientation as a result of the TEM grid preparation. The diffraction spots are localized on the rings with the d-spacing characteristic of the WSe$_2$ material. Indeed, the two rings can be indexed by the (001) and (008) planes, with a d-spacing of 0.284 and 0.162 nm, respectively.

The XRD pattern of the WSe$_2$ sample prepared on sapphire (Figure 3) reveals the presence of very intense peaks at 13.6°, 41.6°, and 56.7°, in agreement with the (002), (006), and (008) diffraction planes of the hexagonal WSe$_2$, respectively (space group P6$_3$/mmc, Joint Committee on Power Diffractions Standards (JCPDS) card (no. 38-1388)). The strong peak intensities with a narrow full width at half

maximum (FWHM) indicate that the sample is highly crystalline (the crystallite size was estimated to be higher than 400 nm using the Debye–Scherrer equation, namely, out of the range where this equation remains valid), and that the nanosheets with van der Waals planes perpendicular to the substrate surface (with a C-axis parallel to the surface) are present [14,29], corroborating the observations made by SEM.

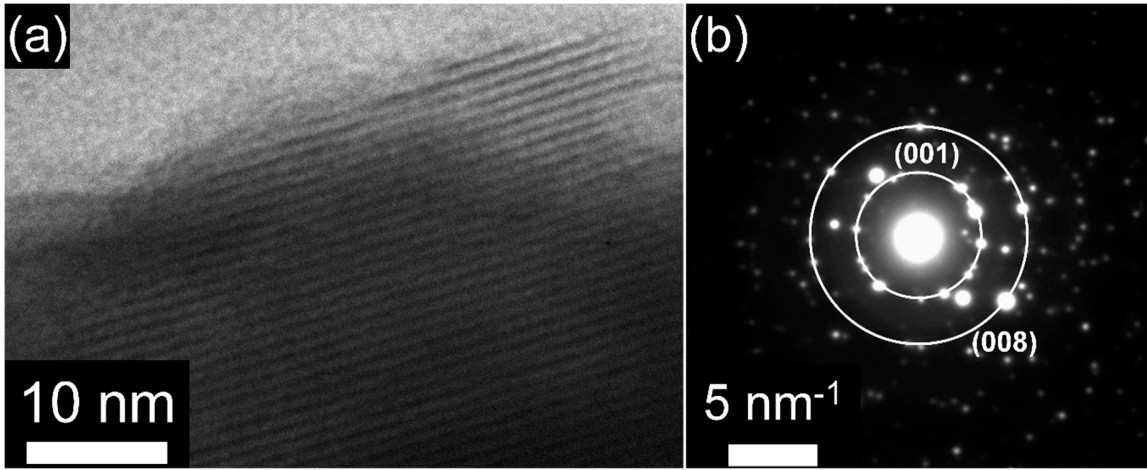

**Figure 2.** TEM images of a (**a**) nanosheet edge and of the (**b**) Selected Area Electron Diffraction (SAED) pattern for $WSe_2$ nanosheets.

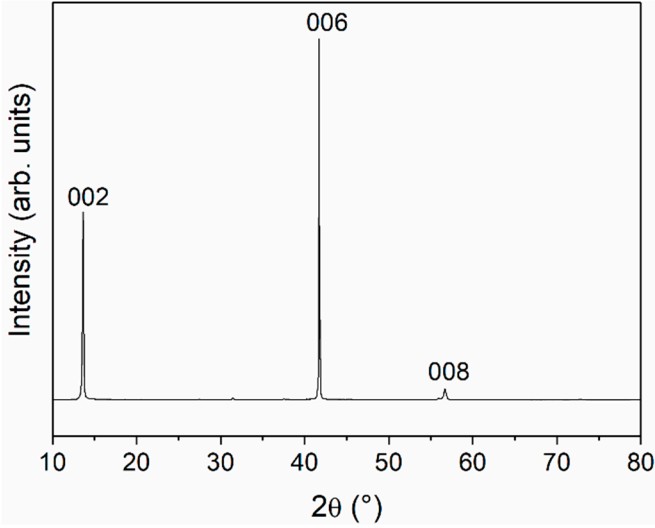

**Figure 3.** XRD pattern of $WSe_2$.

Raman spectroscopy and photoluminescence (PL) are important tools to reveal the optical properties of TMD materials, as well as the number of layers composing the samples [30]. On a typical Raman spectrum of $WSe_2$, recorded with an argon laser excited at a 532 nm wavelength (Figure 4), two unresolved peaks at 251 and 257.6 cm$^{-1}$ are well-visible [30]. The observed peak at 251 cm$^{-1}$ corresponds to the $E_{2g}^1$ mode (in-plane vibrational mode). The shoulder peak observed at 257.6 cm$^{-1}$, associated with the out-of-plane mode, is the $A_{1g}$ mode. This means that a substantial vibration along the vertical layer direction exists (bond vibration between W and Se) [30]. The presence of a Raman peak at 306 cm$^{-1}$ ($B_{2g}^1$ mode) has been reported to be related to the interlayer interaction [22,24]. The Raman signature of the as-grown $WSe_2$ indicates that its structure is a multilayer 2H type.

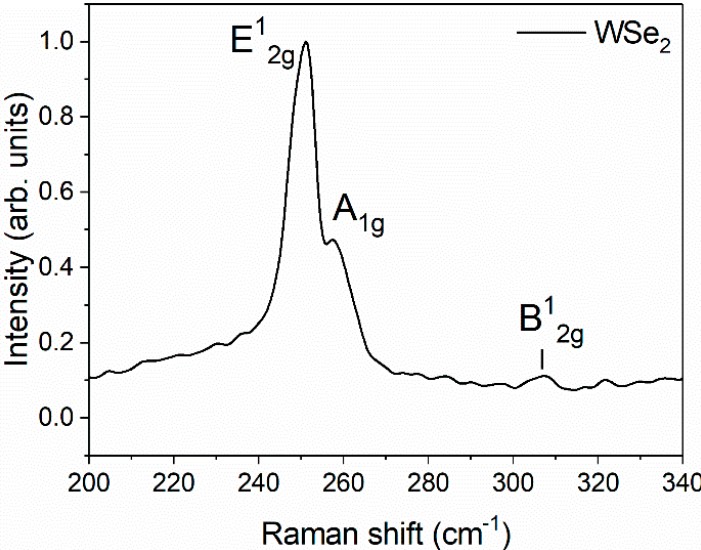

**Figure 4.** Raman spectrum of $WSe_2$.

The PL spectrum of $WSe_2$ is shown in Figure 5. Because of the semiconductor character of the monolayer $WSe_2$, the PL spectrum exhibits a strong direct transition (DT) emission of around 760 nm, due to its direct band gap [31]. When increasing the number of layers of the $WSe_2$, indirect transitions (IT) emissions also appear at a lower energy. The peak of the IT does not appear in the monolayer PL spectrum. Figure 5 shows a main peak at an 860 nm wavelength, which corresponds to a red-shift from 760 to 860 nm, indicating the indirect nature of the transitions. Furthermore, this feature—the presence of IT emissions with important wavelength shifts—demonstrates a multilayer nature, according to the literature. The value of this shift can be considered as a layer number indicator [31].

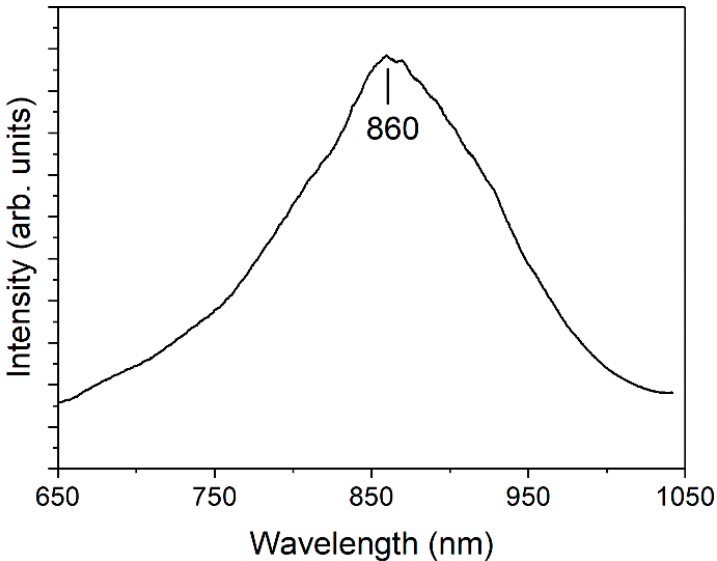

**Figure 5.** Photoluminescence spectrum of $WSe_2$.

Finally, the last optical tool used to characterize the $WSe_2$ sample was ultraviolet–visible spectroscopy. The absorption spectrum is shown in Figure 6. The excitonic absorption peaks of A and B are located at wavelengths of 785 and 651 nm, respectively. These appear because of the direct transitions gap of point K. The presence of excitonic transition, red-shifted from the monolayer transition [32], demonstrates the fewer-layer characteristic of the synthesized nanosheets, according to the PL results. As s result of the superpositions of the Se p-orbitals with W d-orbitals, as well as the

adjacent layers, the $WSe_2$ spectrum shows other absorption peaks of A' and B'. The excitonic nature of these peaks could come from the splitting of the ground and the excited states of the two transitions of A and B, because the d-electron band is perturbed at the level of the inter- and intra-layer by the Se p-orbitals [32,33].

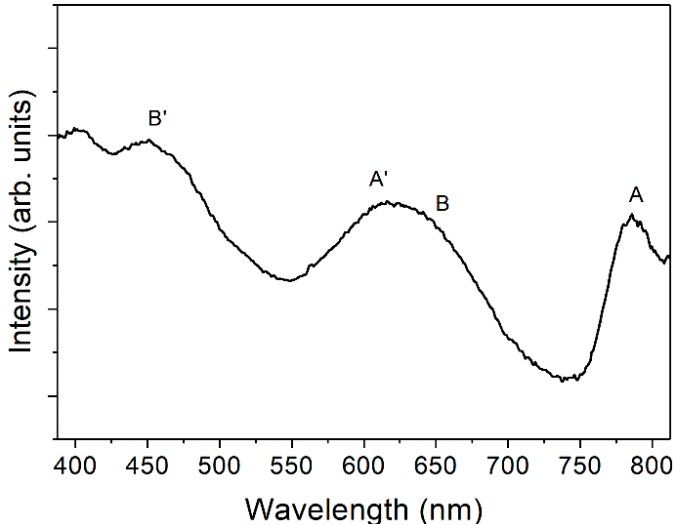

**Figure 6.** UV–VIS spectrum of $WSe_2$.

All of the experimental data obtained with Raman spectroscopy, PL, and UV–VIS spectroscopy, allows for concluding that $WSe_2$ materials are composed of multilayer 2H-$WSe_2$ nanosheets.

The $WSe_2$ nanosheets were investigated by X-ray photoelectron spectroscopy (XPS). The XPS spectra recorded on the W 4f and Se 3d binding energy regions are shown in Figure 7. Considering the W 4f spectrum (Figure 7a), the following three main peaks can be observed: two main contributions at about 32.5 and 34.6 eV are attributed to the doublets W $4f_{7/2}$ and W $4f_{5/2}$, respectively, and a wide and low intense contribution of around 37.8 eV is attributed to the W $5p_{3/2}$. The main contribution can be fitted with two doublets centered at 31.9 and 32.5 eV on a W $4f_{7/2}$ signal, attributed to the 1T' [34] or 1T [35] phase and 2H phase, respectively. The 2H contribution (centered at 32.5 eV) corresponds to 75.0% of the total area, indicating that the sample mainly has a 2H phase. Another additional smaller contribution is observed at 35.5 eV, due to the W 4f loss.

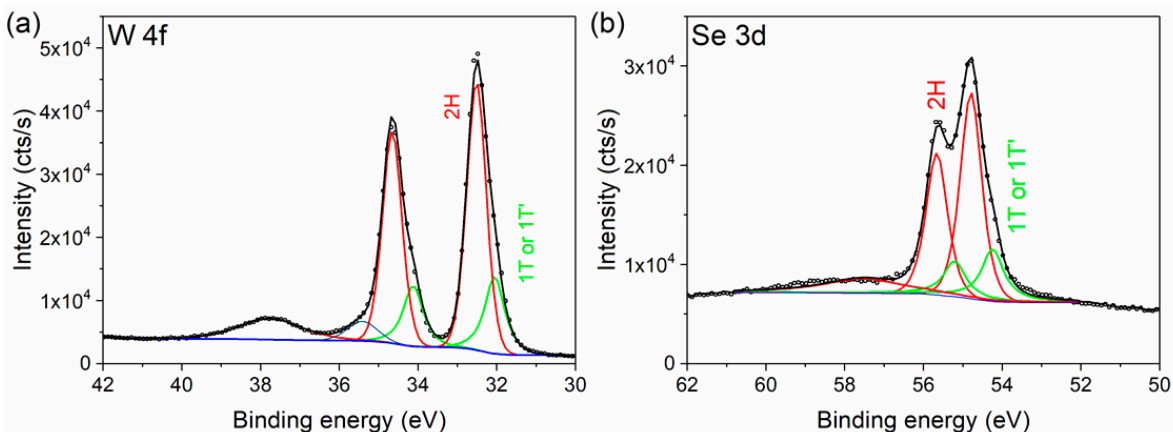

**Figure 7.** XPS spectra of the (**a**) W 4f and (**b**) Se 3d of the $WSe_2$ nanosheets.

The formation of two phases is further confirmed on the Se 3d core level spectra (Figure 7b). The spectrum shows the presence of two main peaks, Se $3d_{5/2}$ and Se $3d_{3/2}$ doublets, at around 54.8 and 55.6 eV, respectively. Each doublet can be fitted with two components—one at 54.8 eV, corresponding

to the 2H-WSe$_2$ phase, and one corresponding to the other phase (1T or 1T'), centered at 54.3 eV. Here, again, the 2H phase represents nearly 75% of the total area (Table 1), verifying the conclusions obtained about the W 4f region. Similar results have been reported in the literature regarding WSe$_2$ crystals [34,35], with a major 2H phase and a minor other phase assigned to the 1T' [34] or 1T [35] phases, respectively.

**Table 1.** Percentages of the corresponding areas of different phases of WSe$_2$ nanosheets.

|  | W 4f Core Level | | Se 3d Core Level | |
| --- | --- | --- | --- | --- |
|  | **2H Phase** | **1T or 1T′ Phase** | **2H Phase** | **1T or 1T′ Phase** |
| **% Area** | 74.3 | 25.7 | 75.0 | 25.0 |

Information about the phases and the quantitative proportion between the phases can be obtained using high resolution XPS spectra, with an energy shift of around 0.8–1 eV in binding energy for the 2H phase compared with the 1T phase, in the two regions of interest, W4f and Se3d [35], and as reported for MoS$_2$ [36]. However, if the distinction between the 2H phase and the 1T phase can be done by XPS, the type of 1T phase—1T or distorted 1T, called 1T'—cannot be elucidated by this technique (only by advanced electron microscopy techniques).

Based on these XPS results, the presence of two phases has been demonstrated in the WSe$_2$ synthesized material.

We must note that Raman spectroscopy can be also a powerful tool to differentiate the number of layers and phases. The MoS$_2$ material can be given in the example where the frequency difference of the two modes, $E_{2g}^1$ and $A_{1g}$, is a function of the layer thickness [37], and the apparition of additional weak modes, $J_1$, $J_2$, and $J_3$, with a decreasing of $E_{2g}^1$ peak intensity for the 1T phase [36,38]. Concerning the WSe$_2$ material, the Raman spectrum of the 1T' WSe$_2$ phase clearly shows a distinct number of weak peaks, which are reminiscent of these J modes [34]. In our studies, however, the contribution of the 2H phase in the Raman signature is overriding, explaining the shape of the spectrum with the high intensity of $E_{2g}^1$. Concerning the results obtained by the photoluminescence and optical absorption spectroscopy, the 1T phases do not produce characteristic features, because of their metallic properties [34,39], and the spectra shape in both cases comes from the semiconducting 2H phase. The last comment concerns the XRD results, where the experimental identification of the different phases is also difficult, because of the similar lattice constants and symmetries.

## 4. Conclusions

The synthesis of the WSe$_2$ film was achieved using an atmospheric pressure CVD technique. We have shown that the synthesized film is constituted of highly crystalline vertically aligned nanosheets. The presence of another phase different to the 2H stable phase has been shown, and could open a new route to tune the physical properties by phase engineering, as has already been done for similar materials, such as MoTe$_2$. The underlying growth mechanism must be further studied in order to establish the best parameters so as to control the synthesis of the WSe$_2$ film with different phases. Further studies will be performed aiming at the evaluation of this material active layer in different applications, such as gas sensing and hydrogen evolution reaction catalysis.

**Supplementary Materials:** The following are available online at http://www.mdpi.com/2076-3417/10/3/874/s1, Figure S1: Scheme of WSe$_2$ synthesis.

**Author Contributions:** A.S.-C. and J.-F.C. conceived, and carried out the experiments. S.A. and C.B. contributed to the Raman and PL measurements. E.H. contributed to XPS measurements and analysis. A.S.-C. and J.-F.C. made the analysis of the data. A.S.-C. and J.-F.C. wrote the manuscript. All of the authors provided critical feedback and helped shape the research and analysis. All authors have read and agreed to the published version of the manuscript.

**Funding:** This research work was financed by a grant from the University of Namur, Belgium. J.-F.C. and C.B. are Senior Research Associates and Research Associates of FRS-FNRS (Belgium), respectively.

**Acknowledgments:** The Synthesis, Irradiation, and Analysis of Materials (SIAM), PC$^2$, and MORPH-IM technological platform of UNamur are acknowledged for the XPS, XRD, and electron microscopy measurements.

**Conflicts of Interest:** The authors declare no conflict of interest. The funders had no role in the design of the study; in the collection, analyses, or interpretation of data; in the writing of the manuscript; or in the decision to publish the results.

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
