# Peer review of "Synthesis and Characterization of Highly Crystalline Vertically Aligned WSe2 Nanosheets"

_applsci, doi:10.3390/app10030874_

Round 1
Reviewer 1 Report
In this manuscript the authors report the synthesis and the characterization of tungsten diselenide nanosheets by using chemical vapor deposotion at atmospheric pressure. The topic of this article is of interest to the reader of the Applied Science journal. The characterization techniques are adequate and the results satisfactorly discussed. The sore point is the poor discussion on possible applications of these materials. Some suggestions follow:
The acronyms should not be included in the abstract. Lines 112-113. How was thickness measured or calculated? Figure 1. The magnification should be indicated. Lines 144-149. Bibliographical references are missing from the discussion of some peaks attributionsAuthor Response
Please see the attachment.

Reviewer 2 Report
Manuscript Number: applsci-691255
Title: Synthesis and characterization of highly crystalline vertically aligned WSe2 nanosheets
The authors present some results about the synthesis and characterization of WSe2 thin films by PACVD. The paper is globally well written but some conclusions and analyzes are quite weak or incomplete. A lot of results are presented but at the end something is amiss and the reader is not fully satisfied and convinced. The topic is nevertheless interesting and deserve improvement.
Consequently, I suggest major revision.
General remarks are followed by detailed ones.
General remarks
Space between value and unit are sometime missing (especially before %).
Detailed remarks are listed below:
I suppose that a word inversion occurs at the end of the abstract. As correctly written in the conclusion, it should be ‘gas sensing” and not “sensing gas”… The second sentence is too long and inconsistent. You should stop the fist part after “one of metal atoms”. The second part is not really correct and concise. I would suggest: “The chemical formula MX2, where M is a transition metal (Mo, W, Nb, Ta, etc.) and X is a chalcogenide (S, Se, etc.) is used to describe their 30 stoichiometry.” Cristal description (from “Between the metal and the chalcogenide” to “neighboring sandwich layers [2].” is not clear and hardly understandable at first reading. It is especially true for the sentence linked to ref 1. The author are invited to work this part new, in order to be precise. I am not convinced by the use of “bulk phase”. You should rather simply used “this bulk material”. “atmospheric pressure system technique (avoids the need of complicated vacuum systems)”. I do not really see the interest of the sentence in brackets. Depends on which level of vacuum you need. If you don’t need an ultra-high vacuum (which is apparently the case) you only need a primary pump. It is not really a system that can be defined as “complicated”. 2 Synthesis. What is the reason of the use of “monitored and controlled”? Is “monitored” not enough? Does “controlled” means that the QMB automatically stop the deposition when 50 nm are theoretically obtained? You should also give more details about the sputtering process. At least, the target size (and shape) and the target to substrate distance. “on a quartz oven”. Don’t you yhink that “in” is more accurate? “(total 0.700 g)”. indeed… you have previously written “2 x 0.350 g” “H2 flow was inserted to help with the reaction”. What is the mentioned reaction? Could you discuss a little more the interest and potential disadvantages of the use of H2? I suggest strongly to work again the synthesis part, which remains rally unclear even after careful reading. A Scheme could be a good idea to help the reader understanding all the steps involved in the process. Results and discussion. Fig 1. What mean “densely grown”? apparently there are domains were the WSe2 grow “vertically” and with a “dense” occurrence of nanosheets. These domains are surrounded by domains with less vertical nanosheets. Why is the grow not homogeneous? What is under the nanosheets (vertical or horizontal)? A stack of horizontal nanosheets? A remaining W layer? Only the sapphire substrate? “nanosheet with a large edge”? what does that mean? Fig 2a. (a and b are missing on the figure). It is hard to understand what is on the picture. Are the black lines, an atomic layer (W or Se) or only W? something is not consistent with the d-spacing. (001) is associated to 0.284, (002) with 0.648 and (008) 0.162. how can d002 be greater than d001? According to ref 2 0.7 nm is the distance between 2 WSe2 sandwich layers. and where is the (002) ring. This picture is not the obvious for the analyze. “crystallites size”. 400 nm is apparently (fig. 1) the lateral dimension of a nanosheet. This should confirm the vertical alignment of the nanosheet. But, I had expected that a nanosheet present a c-axis parallel to the normal of the sheet. It was also consistent with the hexagonal shape observed on fig. 1. You claim the opposite and due to the XRD system used I must agree with this conclusion. But it is not in agreement with Fig1. Or as asked in a previous comment: what is under the vertical nanosheets? If it is horizontal nanosheet it works. Are you sure of the W thickness? You should work this part again and provide more explanations to support your conclusions. I am sorry to ask this question, but I am not speciatist of this part. What is a “2H-WSe2 type”? sheet with 2 sandwich layer? Seems not consistent with the previous SEM observation… “which is large enough 162 for being a layer number indicator” and what is then the layer number? “demonstrates the few-170 layer characteristic of the synthesized nanosheets in agreement with the PL result” ? Nothing quantified is given. How can we follow you? Same question about “1T”. I have to suppose it is Tetragonal. And? You should explain a little what the presence of this phase implies. The conclusion must be update after correction of the text.
Reviewer 3 Report
Review of “Synthesis and Characterization of Highly Crystalline Vertically Aligned WSe2 Nanosheets”, by Ayrton Sierra-Castillo et al.
Dear authors,
I congratulate you for the amount of work presented in this manuscript.
Although the manuscript reports an interesting subject, it should be re-written again such that the consistency between the presented materials is improved. By the current shape, it is hard to follow your manuscript. Therefore I recommend a major revision.
You will find in the following my detailed comments:
Line 42-44: what do you mean by WSe2? Monolayer? Nanosheet?“Moreover, WSe2 possesses a smaller band-gap than monolayer MoS2 and it exhibits ambipolar transport phenomenon [8].”
Line 79-80: what is Ar flow rate unit? Please, correct. Line 81-84: how do you know the temperature of the different zones in the CVD device? What is the brand and model of the CVD device you used? Line 82-83: how have you chosen the amount of Se powder in the CVD furnace? Based on the W size, weight, …? Please explain. Line 85-90: the two-step selenization is not clear. Please re-write these lines and if it is possible use schematic images to show the experimental setup. Line 159-160: “As the number of layers of the WSe2 increases, indirect transitions (IT) emissions show up in lower energy side the DT ones.” Could you please explain the reason behind this phenomenon?
Best wishes,
Reviewer

Round 2
Reviewer 2 Report
Manuscript Number: applsci-691255-revised1
Title: Synthesis and characterization of highly crystalline vertically aligned WSe2 nanosheets
The authors present their revised paper about the synthesis and characterization of WSe2 thin films by PACVD. Following the reviewer recommendations, the paper was improved significantly and is yet ready for publication.
Deeper analyzes and comparison could be still awaited, but I have to suppose this will be performed in a future paper.
Few minor corrections (listed below) must be performed before publication without the need of a third review.
“X is a chalcogenide (X=S, Se, etc.)”: “X=” can be removed from the brackets. “A pure W target of 57 mm was used”. You should ad “in diameter” after “57 mm” to be completely precise. “The samples were characterized by electrons microscopies:” the sentence can be removed. The paragraph can without any understanding problem start with “Scanning and transmission …”. Line 162 “the nature indirect nature of transitions” the first “nature” has to be removed
Reviewer 3 Report
Dear authors,
The authors have seriously and carefully responded to the questions after the first manuscript.
I have two comments for the authors that you will find them in the following:
Line 78-80: please correct the Ar flow unit. “The reactor was flushed for one hour to remove (reduce) oxygen using a 475 l mn−1 Argon flow outside of the furnace, prior to the selenization process.” Line 183-204: I would recommend you to show the name of different peaks on the XPS image "Fig. 7" and prepare a Table with a different peak area. Since in the lines 195-196 you have presented the peak area assign the 2H phase: "Here again, the 2H phase represents near 75% of the total area, verifying the conclusions obtained on the W 4f region.", a Table can show the peak area of the different peaks and it would be useful for the reader.
Best wishes,
Reviewer
